# Mice as an Animal Model for Japanese Encephalitis Virus Research: Mouse Susceptibility, Infection Route, and Viral Pathogenesis

**DOI:** 10.3390/pathogens12050715

**Published:** 2023-05-14

**Authors:** Jordan C. Frank, Byung-Hak Song, Young-Min Lee

**Affiliations:** Department of Animal, Dairy, and Veterinary Sciences, College of Agriculture and Applied Sciences, Utah State University, Logan, UT 84322, USA; jcfrank.v@gmail.com (J.C.F.);

**Keywords:** Japanese encephalitis virus, flavivirus, animal model, mouse susceptibility, infection route, viral pathogenesis

## Abstract

Japanese encephalitis virus (JEV), a zoonotic flavivirus, is principally transmitted by hematophagous mosquitoes, continually between susceptible animals and incidentally from those animals to humans. For almost a century since its discovery, JEV was geographically confined to the Asia-Pacific region with recurrent sizable outbreaks involving wildlife, livestock, and people. However, over the past decade, it has been detected for the first time in Europe (Italy) and Africa (Angola) but has yet to cause any recognizable outbreaks in humans. JEV infection leads to a broad spectrum of clinical outcomes, ranging from asymptomatic conditions to self-limiting febrile illnesses to life-threatening neurological complications, particularly Japanese encephalitis (JE). No clinically proven antiviral drugs are available to treat the development and progression of JE. There are, however, several live and killed vaccines that have been commercialized to prevent the infection and transmission of JEV, yet this virus remains the main cause of acute encephalitis syndrome with high morbidity and mortality among children in the endemic regions. Therefore, significant research efforts have been directed toward understanding the neuropathogenesis of JE to facilitate the development of effective treatments for the disease. Thus far, multiple laboratory animal models have been established for the study of JEV infection. In this review, we focus on mice, the most extensively used animal model for JEV research, and summarize the major findings on mouse susceptibility, infection route, and viral pathogenesis reported in the past and present, and discuss some unanswered key questions for future studies.

## 1. Introduction 

Japanese encephalitis virus (JEV) is a mosquito-borne arbovirus taxonomically belonging to the genus *Flavivirus* in the family *Flaviviridae* [1]. Of the 53 classified species of known flaviviruses, JEV is related to other medically important mosquito-borne flaviviruses to various genetic and antigenic extents, with a much closer relationship being noted with West Nile (WNV), St. Louis encephalitis (SLEV), and Murray Valley encephalitis viruses than with Zika (ZIKV), dengue (DENV), and yellow fever (YFV) viruses [2,3]. JEV circulates in nature by both horizontal transmission among susceptible animal hosts, primarily through the bite of culicine mosquito vectors (e.g., *Culex tritaeniorhynchus*) [4,5], and vertical transmission from female mosquitoes to their offspring through the transovarial infection of developing eggs [6,7]. Of many susceptible animals, certain vertebrates, such as suids (e.g., domestic pigs) and avians (e.g., wading birds), are especially relevant for the incidental transmission of JEV to the human population [8,9,10]. Experimentally, JEV can also be transmitted in the absence of mosquito vectors through several non-vector-borne routes in animals such as rodents, pigs, bats, and/or squirrel monkeys: contact transmission [11,12,13,14], aerosol transmission [11,15], transplacental transmission [16,17,18,19,20], and artificial insemination [21,22]. In both humans and animals, JEV causes Japanese encephalitis (JE), formerly called Type B or Japanese B encephalitis [5,23], which is an acute encephalitis syndrome that potentially leads to severe brain damage or death [24,25]. In Japan, where JE was first recognized, the disease is believed to have arisen in the “summer encephalitis” season before the 20th century began, although the first recorded outbreak occurred in 1924, involving >6000 human cases with a case fatality of ~60% [26,27]. About a decade after this historic outbreak, a filterable agent from the brain of a fatal human JE case was demonstrated to be able to produce the disease in monkeys. In 1935, JEV was first isolated from the brain of a deceased patient, and the isolate was designated as the Nakayama strain [28]. Despite the fact that JEV has been known as the cause of JE for nearly a century, it is still a neglected pathogen that continues to be a major public health challenge [29]. 

JEV is the most common cause of viral encephalitis in the Asia-Pacific region [30,31,32,33], with the reported boundaries of viral activity on the north being much of China and the eastern China-Russia borderlands [34,35,36,37,38,39,40], on the south being Papua New Guinea [41,42] and a northern part of Australia [43,44,45,46,47,48,49], on the east being Guam [50,51,52] and Saipan [53,54], and on the west being much of India [55,56] and a southeastern part of Pakistan [57,58]. This region contains ~25 countries, including the top two most populous countries (i.e., China and India) of the world and several of the most densely populated countries (e.g., Singapore and Bangladesh), putting far more than half of the global population at risk of JEV infection [59]. Additionally, some cases of JEV infection were found in birds [60,61] and mosquitoes [62] collected in Italy during the first decade of the 21st century, which was the first time that the virus was detected outside the Asia-Pacific region, albeit with no human JE outbreaks reported as of yet. In 2016, however, a case of co-infection with JEV and YFV was identified unexpectedly in an Angolan resident with no out-of-country travel history [63]. This recent epidemiological data indicates that JEV is no longer confined to the Asia-Pacific region, and is continuously expanding its activity over new territories in Europe and Africa, with possible spread to the Americas in the very near future [64,65]. Hence, JEV is an emerging pathogen that has the potential to spread across the globe, now more than ever [66,67,68]. 

The annual incidence of JE presents two main epidemiological patterns [69,70], intrinsically aligned with the regional climate that affects the population size, density, and distribution of both mosquito vectors and animal hosts involved in JEV transmission [71,72]. In tropical regions, JEV is “endemic,” causing sporadic outbreaks nearly all year round at low frequencies, with peaks in the rainy season affecting all young under age 15 due to lack of immunity to the virus. In subtropical and temperate regions, on the other hand, JEV is “epidemic”, causing seasonal outbreaks exclusively during the summer and early fall and affecting all ages, with a bimodal distribution peaking in young children and the elderly because of the lack and the fading of JEV-specific immunity, respectively. Since its emergence, JEV has evolved into five divergent genotypes (G1–G5) presumably at the center of the Asia-Pacific region (i.e., Indonesia and Malaysia), from which one or more genotypes have dispersed to other areas of the region [73]. Historically, G1, G2, and G3 have all been circulated broadly at various times and frequencies in the Asia-Pacific region, except the India-Nepal-Sri Lanka area in which only G1 and G3 have been detected, and the Australia-Papua New Guinea area in which only G1 and G2 have been found [74,75,76,77,78,79,80,81,82,83]. Of particular note, G3 was the most prevalent in Asia until the 1990s, but since then, it has slowly been replaced by G1 [74,75,76,77,78,79,80]; yet, G3 has recently appeared for the first time outside the Asia-Pacific region in Italy [60,61,62] and Angola [63]. Unlike the three aforementioned genotypes, G4 has been located only in the Indonesia-Malaysia area [84], and G5 has been limited to the China-Japan-Korea area apart from the Indonesia-Malaysia area [85,86,87,88]. Within the five genotypes, it is noted that the sequence divergence between their genomes at the nucleotide (nt) and amino acid (aa) levels reaches up to ~20% and ~10%, respectively [85]; however, their biological differences in viral transmissibility, pathogenicity, and immunogenicity have not yet been fully understood. 

From a clinical standpoint, JEV causes mostly asymptomatic infections, with the symptomatic cases in humans ranging from ~0.1% to <5.0%, depending on the study population evaluated [89,90]. Despite such a low symptomatic infection rate, the annual number of new JE cases is estimated to be ~68,000 with a case-fatality rate reaching up to ~30% and affecting predominantly children under the age of 15 [91,92]. Furthermore, up to ~50% of JE survivors may never fully recover and continue to seek medical care for long-term neurologic, psychiatric, cognitive, and/or behavioral problems with various degrees of severity [93,94,95]. As interventions for the control of JEV, one or more different types of vaccines are commercially available in certain parts of the world [96,97,98], yet no antivirals have been developed [24]. Even with readily available vaccines, ~80% of new annual JE cases still occur in countries where national immunization programs are already in place or under development [92,99]. This data, and the fact that JEV is a vector-borne zoonotic pathogen, underscores the unmet need to develop therapeutics for the treatment of JEV infection. In order to facilitate the development of such drugs, we herein review the previous studies that were conducted using various strains of mice, the most widely accepted laboratory animal model for the study of JEV replication and JE neuropathogenesis. 

## 2. The Virus 

JEV is an enveloped virus with a linear, single-stranded, positive-sense RNA genome [100]. The virion consists of an inner nucleocapsid composed of the genomic RNA and multiple capsid C:C homodimers [101], and an outer shell composed of two surface proteins M and E arranged into 90 E:M:M:E heterotetramers [102] in a herringbone pattern on the viral envelope membrane (Figure 1, top panel). The genomic RNA is ~10,977 nt long (Figure 1, bottom panel) and has three functional parts [103]: (1) an ~95-nt untranslated region (UTR) with a type 1 cap structure attached at the 5’ end [104], (2) an ~10,299-nt central open reading frame (ORF) with a -1 ribosomal frameshift signal located at positions nt 3551-3630 [105], and (3) an ~583-nt UTR with no poly(A) tail at the 3’ end [106]. Each of the 5’ and 3’ UTRs, as well as the ~100-nt 5’-terminal region of the ORF, contains a network of cis-acting RNA elements [107,108,109] defined by the primary, secondary, and tertiary structures that regulate the translation and replication of the viral genomic RNA [110,111,112,113]. The single long ORF encodes two precursor polyproteins [114]: (*i*) the full-length ppC–NS5, which undergoes co- and post-translational proteolytic cleavages to yield three structural (C, prM/M, and E) and seven nonstructural (NS1, NS2A, NS2B, NS3, NS4A, NS4B, and NS5) proteins; and (*ii*) the frameshift-derived truncated ppC–NS1’, which also goes through the same proteolysis as does ppC–NS5 to produce the three aforementioned structural proteins and NS1’, a 52-aa longer isoform of NS1 as a result of its C-terminal extension that includes the 9-aa N-terminal segment of NS2A before the frameshift and a 43-aa unique peptide synthesized from the new reading frame afterwards [103,115,116]. Of these 10 mature proteins, the following two are enzymes, each containing multiple catalytic activities [117,118,119,120]: (*a*) NS3, 619 aa in length, has serine protease activity in the ~168-aa N-terminal region [121] and helicase, nucleoside 5’-triphosphatase, and RNA 5’-triphosphatase activities in the ~438-aa C-terminal region [122], and (*b*) NS5, 905 aa in length, has methyltransferase and guanylyltransferase activities in the ~265-aa N-terminal region and RNA-dependent RNA polymerase activity in the ~630-aa C-terminal region [123]. In a nutshell, the three structural proteins are required for the formation of an infectious virion [124,125,126], and the seven nonstructural proteins are engaged in multiple steps of viral replication (i.e., polyprotein processing [127], RNA replication [128,129,130], and particle morphogenesis [131,132]) and in a variety of host cell responses to viral replication [133,134,135,136]. With regard to JEV pathogenesis, the viral E protein has been the target of extensive study since it acts as the viral receptor interacting with an ill-defined cellular component(s) on susceptible host cells for viral entry [137,138,139]. In addition, a recent study has suggested that a secreted form of the viral NS1 protein specifically binds to brain endothelial cells and alters their permeability, causing brain-specific vascular leakage that contributes to the pathogenesis of JE [140]. 

## 3. Clinical Features in Humans

For symptomatic JEV infection, it generally takes 5–15 days with a median of 8.4 days to show clinical signs/symptoms after contraction of the virus, but in some cases, may have a longer incubation time of 3–4 weeks [141]. The clinical outcomes vary significantly [90,142] from mild self-limiting undifferentiated illnesses (e.g., fever) to severe life-threatening nervous system diseases (e.g., encephalitis) [143,144,145,146]. 

### 3.1. Central Nervous System Disorders

JE, a neuroinflammatory disease, is the most serious outcome of JEV infection attacking neurons and other cell types in the central nervous system (CNS) [24]. JE almost always starts with nonspecific symptoms, such as lethargy, fever, chills, coryza, and diarrhea for a few days prior to the onset of neurological manifestations, such as headache, vomiting, impaired consciousness, altered mental status, and seizures [147,148,149,150,151,152,153,154,155]. Seizures tend to occur more frequently in young children [151,152,156]. Critically ill JE patients may present a Parkinsonian syndrome with expressionless faces, tremors, and hypertonia linked to the damage of the basal ganglia in the brain [157,158] or a polio-like syndrome with acute flaccid paralysis caused by destruction of the anterior horn cells in the spinal cord [159,160,161,162,163,164]. Other movement disorders observed in JE patients include opisthotonus, dystonia, hemiparesis, choreoathetosis, orofacial dyskinesias, and gaze palsy [148,149,150,153,165]. Occasionally, JE patients develop neuroimmunological complications, such as anti-N-methyl-D-aspartate receptor (NMDAR) encephalitis caused by an antibody-mediated autoimmune reaction to the NMDARs in the brain [166,167,168], acute transverse myelitis characterized by an inflammation in the spinal cord often damaging the myelin sheath [169,170], and acute disseminated encephalomyelitis noted in general by an inflammatory demyelination affecting the CNS [171,172]. In addition, non-neurological manifestations are also associated with JE, which include gastric hemorrhage, thrombocytopenia, liver dysfunction, hepatomegaly, splenomegaly, and potentially pulmonary edema [148,173,174]. Poor prognosis for JE is indicated by persistent/recurrent fever, prolonged/repeated seizures, poor plantar reflexes, severe lung and kidney problems, high JEV antigens in cerebrospinal fluid (CSF), and low anti-JEV IgM antibodies in CSF and serum [151,153,175,176,177,178]. Even after recovery, a large portion of JE survivors have neuropsychiatric sequelae, such as memory loss, intellectual disability, language disorders, emotional disorders, behavioral disorders, movement disorders, and ocular motor disorders [179,180,181,182,183]. 

### 3.2. Peripheral Nervous System Disorders

Limited clinical studies have suggested that JEV infection may induce Guillain–Barré syndrome (GBS) [184,185,186], a neuroimmunological disease caused by our own immune system primarily marring the myelin sheaths and axons of nerves of the peripheral nervous system (PNS) [187]. To date, the most informative study conducted with 47 JEV-positive GBS patients has shown that the disease usually begins with febrile illness, as seen during the prodromal phase of JE, which then progresses to neurological complications, such as headache, disturbed consciousness, paresthesia, impaired reflexes, flaccid paralysis, and breathing problems [185]. In the same study, a follow-up examination with 17 of the initial 47 GBS patients at eight months after hospital discharge found that they still experienced neuromuscular complications, such as limb muscle weakness and atrophy, incontinence, and hoarse voice [185]. Although its causal relationship with GBS requires further investigation, JEV appears to be able to trigger the PNS-damaging GBS. In line with this notion, other flaviviruses have recently been documented to be a cause of GBS, which include WNV [188,189], ZIKV [190,191], and DENV [192,193]. 

### 3.3. Reproductive Disorders

It is conceivable that pregnant women infected with JEV may pass the virus to their fetus, based on the recent findings that during gestation, ZIKV can cross the placenta, infect the unborn offspring, and thus cause miscarriage or a variety of birth defects, collectively designated as congenital Zika syndrome, such as microcephaly [194]. However, little is known about the possible adverse effects of JEV infection during pregnancy. Until now, only a few human cases of congenital JEV infection have been reported, which result in either abortions or the deliveries of a healthy child [195,196,197,198], possibly depending on the time of infection during gestation [197]. In pregnant sows, on the other hand, it has been well established that congenital JEV infection causes reproductive losses, such as mummified, stillborn, and abnormal piglets even when they are born alive [199,200,201,202,203]. Therefore, there is a clear difference in the clinical outcome of congenital JEV infections between humans and pigs, both of which are highly susceptible to the virus. It may be worth investigating a possible causal link between JEV infection and pregnancy complications in humans, and if there is a link, then its potential effects on fetal growth in general and fetal brain development in particular, should be studied. 

## 4. Mice: A Reliable Small-Animal Model for JEV Research

Mice are a well-characterized and widely accepted animal model for JEV research, studying viral infection and disease pathogenesis, as well as evaluating the safety and efficacy of potential medical countermeasures at a preclinical stage of development. The mouse model has many advantages over other animal models, of which the four most compelling are [204]: (1) the biological properties, such as small body size, rapid reproduction cycle, and short life span are favorable for handling and husbandry in a high-level biosafety laboratory; (2) the well-characterized CNS enables us to extrapolate new findings to the human CNS; (3) biological reagents are available for a variety of cell biological and immunological analyses; and (4) an increasing number of genome-editing technologies are applicable for the extension of future genetic studies. However, there are also some limitations due to the fact that there exist a range of genetic, anatomical, physiological, and immunological differences between mice and humans [205,206], which reflects how well mice recapitulate key biological processes contributing to the development and progression of JEV-induced diseases. Indeed, a systematic comparative study has shown that, in some human inflammatory conditions, the gene expression profiles are not reproduced in their mouse models [207]. Moreover, in cancer studies, it has been indicated that the process of tumorigenesis is fundamentally different in humans and mice [208]. Nevertheless, to date, mice have been the most commonly used animal model to investigate the genetic, molecular, and cellular basis of JEV replication and JE neuropathogenesis [209,210,211]. 

### 4.1. Mouse Susceptibility 

Mice vary in their susceptibility to JEV infection (Figure 2), depending on a combination of the viral strain, inoculum dose, and inoculation route on one side [212,213], and the mouse strain and age on the other side [213,214,215]. 

#### 4.1.1. Mouse Strain-Dependent Variation

As for the importance of host genetic background in determining its susceptibility to JEV infection, an earlier study using eight different strains of adult mice (70–120 days old) has described three classes of mouse susceptibility to peripheral infection with a given equal dose of a wild-type (WT) JEV strain, based on their infection rates (IRs) and mortality rates (MRs): (1) high IRs and high MRs (e.g., C3H/He), (2) high IRs and medium MRs (e.g., C57BL/6), and (3) low IRs and low MRs (e.g., BALB/c) [214]. In the same study, a closer comparison of C3H/He with C57BL/6 mice has revealed that JEV replicates in various internal organs, produces viremia in the blood, and even penetrates into the brain equally well in both mouse strains when inoculated intraperitoneally; however, lethal encephalitis has been shown to occur more frequently in C3H/He than in C57BL/6 mice. In contrast, no significant difference in mortality has been observed between C3H/He and C57BL/6 mice when inoculated intracerebrally [214]. Although the number of animals used in the study is small, these data suggest that C57BL/6 may elicit stronger immune responses following peripheral JEV infection than C3H/He, which in turn, suppress viral replication in the brain even if the virus enters the brain [216]. This notion is in line with a recent report, describing that a lower mortality rate observed in DBA/2 mice as compared to C3H/HeN after peripheral JEV infection is related to a more rapid and robust neutralizing antibody response, and also possibly to the lower permissiveness of their macrophages (MΦs) and dendritic cells (DCs) to the infection, which may lead to a decrease in viral neuroinvasion and/or neuropathogenesis even after viral entry into the CNS [217]. As seen with JEV, variation in mouse susceptibility has also been noted with other flaviviruses [218,219,220,221]. Although the genetic basis of this mouse strain-dependent susceptibility is not yet fully understood, a significant but limited number of cellular genes, such as those involved in dsRNA- and interferon (IFN)-mediated innate immune responses, have been shown to affect mouse susceptibility to infection with one or more flaviviruses, such as WNV, ZIKV, DENV, and YFV [222]. For JEV, it has been reported that blockage of CD137 (also called 4-1BB and TNFRSF9), a costimulatory molecule expressed on MΦs, DCs, NK cells, T cells, and B cells, makes C57BL/6 mice less susceptible to infection, by enhancing the innate immune responses involving the rapid activation of type I/II IFN responses and the increased infiltration of mature Ly6C^hi^ monocytes into the inflamed CNS [223]. In addition, a previous study using mice lacking various Toll-like receptors (TLRs), a key element in activation of the innate immune system, has found that TLR3 knockout makes its parental BALB/c mice more susceptible to JEV infection, but interestingly, TLR4 knockout makes its parental C57BL/6 mice less susceptible to JEV infection due to the enhanced type I IFN-mediated innate immune responses, as well as the augmented adaptive humoral and cellular immune responses [224]. Moreover, it has been suggested that, in addition to laboratory mice, wild mice may offer some advantages in identifying cellular genes associated with individual differences in susceptibility to JEV infection within natural populations [225]. 

#### 4.1.2. Mouse Age-Dependent Variation

The age of mice is another key factor as important as their genetic make-up for their susceptibility to JEV infection. A very early study using the DD strain of albino mice has shown that the level of its susceptibility to subcutaneous inoculation of a WT JEV strain with a dose of 18–40 plaque-forming units (PFU) markedly decreases as the mice get older, specifically from being highly susceptible at 1 week of age, displaying 100% mortality, to becoming resistant at 2 weeks of age, exhibiting only 25% mortality [226]. However, those 2-week-old mice were found to be highly susceptible reaching the mortality of 100%, when inoculated with a much higher dose of ~3.2 × 10^5^ PFU through the same subcutaneous route [226]. In addition, our recent study using outbred CD-1 mice, when inoculated intracerebrally with SA_14_-14-2, the live-attenuated JEV vaccine licensed for human use, has demonstrated that the 1-week-old or younger mice are highly susceptible, having an estimated 50% lethal dose (LD_50_) of <1.5 to 3 PFU, but the 2-week-old or older mice are resistant, having an estimated LD_50_ of >1.5 × 10^3^ PFU [103]. One unexpected finding from this study is that SA_14_-14-2, although fully attenuated, is able to replicate in the brain of suckling CD-1 mice (1–7 days old) and causes fatal encephalitis. As expected, however, we also found that the 3-week-old CD-1 mice are fully resistant to both intracerebral and peripheral infections with SA_14_-14-2, presenting LD_50_ values of >1.5 × 10^5^ PFU [103]. These results suggest that, although younger mice (e.g., <1 week old) are more susceptible to JEV infection, a certain age group (e.g., 3–4 weeks old) is better suited for the study of viral pathogenicity, disease pathogenesis, and host immune response, as it better recapitulates the disease development and progression observed in humans. The age-dependent susceptibility to JEV infection has also been reproduced in a rat model, showing that the resistance is developed at the age of 2 weeks [227]. Furthermore, mouse age-dependent susceptibility has been documented for YFV [228]. Given the fact that the vast majority of JEV infections in humans are asymptomatic or self-limited, it is of particular interest to understand the molecular basis of the strain- and age-dependent susceptibility of mice and to identify human counterparts. 

### 4.2. Infection Route 

A natural JEV infection in human and animal hosts occurs principally through an intradermal or subcutaneous route, when they are bitten by an infected mosquito [229]. An experimental JEV infection in mice, on the other hand, can be performed through one of several different inoculation routes (Figure 3), depending on the purpose of the experiments. In some cases, it may not reflect the natural infection, but the virus can still cause fatal encephalitis. 

#### 4.2.1. Peripheral vs. Intracerebral Inoculation 

As a peripheral route of JEV inoculation, footpad injection has been used, albeit rarely, to mimic the natural infection as a mosquito bite, which allows the virus to spread from a primary infection site in a hindlimb footpad to the CNS [230,231], reportedly affecting the thalamus first, then the sensory cortex and other areas of the CNS, and further spreading to parts of the PNS, such as the somatic and autonomic nervous systems [232]. Like in mice, the uni- or bi-lateral thalamic lesions are also recognized in humans at the early phase of JEV neuroinvasion into the CNS [163,233,234]; however, the involvement of the PNS and its clinical significance during the course of JEV infection in humans are less clear, although WNV is reported to be able to infect sensory and motor neurons of the PNS in both rodents [235,236] and humans [237,238]. As compared to footpad injection, other peripheral inoculation routes, especially intramuscular and intraperitoneal injections, have been employed rather frequently because of their methodological convenience, although the exact path of viral neuroinvasion may not be identical to that of the mosquito-mediated natural infection. The peripheral inoculations have been used to understand the dynamic biology underlying JEV pathogenicity overall, in particular neuroinvasiveness, as well as host immune responses to JEV infection [103,217,239,240,241,242,243]. In addition, the intracerebral route of viral inoculation has been utilized specifically to elucidate the complex biology underlying JEV neurovirulence and host neuroinflammatory responses to JEV infection in the brain [244,245,246,247]. 

Regardless of the route of JEV inoculation (peripheral or intracerebral), clinical signs in infected mice typically begin with decreased activity, ruffled fur, and weight loss that gradually progress to hunched posture, impaired coordination, tremors/convulsions, hindlimb paralysis, and death; however, the median survival time after peripheral inoculation is always one to several days longer than that after intracerebral inoculation [103,242,244]. Interestingly, a comparative microscopic study of the brains taken from terminally ill mice after JEV inoculation via the intraperitoneal or intracerebral route has shown that more severe inflammation occurs in the intraperitoneally infected group, as characterized by the infiltration and accumulation of leukocytes around the blood vessels, with mononuclear leukocytes often being in close contact with degenerated and disintegrated neurons, than in the intracerebrally infected group that presents only low levels of glial cell proliferation and mononuclear leukocyte infiltration, although a large number of neurons are infected, as seen in the intraperitoneally infected group [248]. This previous study suggests that JEVs injected directly into the brain infect neurons and induce the cell death that is a direct cause of mortality, whereas JEVs administered into the peritoneal cavity may stimulate the host antiviral immune responses prior to viral neuroinvasion that triggers massive inflammation against JEV-infected neurons in the brain afterwards. In addition, a comparative immunological study of the lymphocytes collected from three different strains of mice after JEV infection through three different routes has found that JEV-specific T-helper (Th)-cell responses, as determined by the relative expression levels of Th1 (IFN-γ) and Th2 (IL-4) cytokines, depend not only on the mouse strain used (i.e., inbred C57BL/6J or BALB/c vs. outbred Swiss albino), but also on the inoculation route employed (i.e., intraperitoneal or subcutaneous vs. peroral) [241]. Overall, these results highlight the differential impact of viral inoculation route on the regional immune responses of a host to JEV infection, which affects the systemic immune responses and ultimately the clinical outcome. 

#### 4.2.2. Inoculation into a Cephalic Cavity

As an animal model for the study of JEV infection, a collection of mouse strains has been shown to be susceptible to the virus by peripheral inoculation almost always at an anatomic site distant from the brain, such as the hindlimb footpad, peritoneal cavity, and thigh muscle, but on some occasions, into a cephalic cavity close to the brain, such as the nasal, oral, and orbital cavities via one of the following inoculation routes: 

(1) *Intranasal/oronasal route*. An early study, in which groups of 3- and 7-week-old CD-1 mice were exposed to aerosols containing a WT JEV strain, has found that the virus causes lethal encephalitis in both age groups, with a higher mortality rate observed in the younger group [15]. Similarly, a recent study, in which 4-week-old BALB/c mice were inoculated intranasally with the virulent SCYA201201 or attenuated SA_14_-14-2 strain of JEV, has shown that both strains replicate well in the lung, liver, heart, kidney, thymus, spleen, and brain [11]. This study has further indicated that both JEV strains are transmissible from the intranasally infected to naïve mice by contact or through aerosols [11]. However, one caveat of this latter study is that none of the mice infected with SCYA201201 or SA_14_-14-2 had any detectable levels of neutralizing antibody or showed any clinical signs of JEV infection, although histopathological lesions were visible in their lungs and brains [11]. Still, these two studies suggest that JEV can establish a systemic infection in mice following intranasal/oronasal inoculation. Similar to mice, pigs are also shown to be susceptible to JEV infection through an oronasal route [13,14,249], as the virus persist in tonsils for at least 3–4 weeks and in oral fluids for up to 2 weeks [12,14,250]. These data suggest that the direct pig-to-pig transmission of JEV may play a role in overwintering of the virus when its mosquito vectors are unavailable [251,252]. Furthermore, JEV is reported to be infectious as inhalable aerosols to several other experimental animals, of which Syrian hamsters and three non-human primates (NHPs; squirrel, cynomolgus, and rhesus monkeys) develop lethal encephalitis, whereas Fisher–Dunning rats and Hartley guinea pigs exhibit only asymptomatic infection even with a high inoculum of up to 10^5^ PFU [15,253,254]. Other flaviviruses are also documented to be transmissible via intranasal, oronasal, or other mucocutaneous routes in various animal species and/or humans (e.g., WNV in birds, mice, hamsters, monkeys, and humans [255,256,257,258,259]; ZIKV in guinea pigs, monkeys, and humans [260,261]; SLEV in mice [262]; DENV in humans [263]; Wesselsbron virus in mice and humans [264]; Bagaza virus in birds [265]; and Tembusu virus in ducks [266]). 

(2) *Peroral route*. A previous study, in which young adult Swiss albino mice were inoculated perorally, intraperitoneally, or subcutaneously with the JEV strain P20778 that fails to cause encephalitis in weanling or adult mice after peripheral inoculation, has found that the perorally infected mice generate protective antibodies directed against JEV proteins (e.g., E and NS1), although the antibody titers rise in a much slower rate as the number of inoculation increases as compared to those produced in mice infected intraperitoneally or subcutaneously [267]. In addition to the antibody response, it has also been noted that the perorally infected mice develop JEV-specific T-helper cell cytokine responses, although it is less clear which subset of T-helper cells are induced (Th1 or Th2), but they are distinct from those produced in the mice infected intraperitoneally or subcutaneously that uniformly elicit a strong Th1 response [241]. Even though these studies do not provide direct evidence for the occurrence of viral replication in the perorally infected mice, they seem to suggest a remote, yet distinct, possibility of mice being susceptible to peroral JEV infection. 

(3) *Ocular route*. A very recent study, in which 2-week-old Swiss albino mice were inoculated via a conjunctival route with a WT JEV strain, has shown that this ocular inoculation produces a systemic infection, determined by clinical signs of infection with 100% mortality, histopathological lesions in the brain, and viral RNAs/antigens in various visceral organs and many parts of the brain, as well as in the neuronal cells of the eye [268]. In agreement with this finding, a very early report, in which 3-week-old mice of an unspecified strain were infected with a WT JEV strain through a conjunctival, intrabulbar or retrobulbar route, has also found that all three ocular routes of inoculation lead to lethal encephalitis [269]. In the same report, it was further shown that the intrabulbar inoculation leads to a shorter average mouse survival time than the intranasal inoculation, but longer than the intracerebral inoculation [269]. 

In summary, multiple strains of both outbred and inbred mice are shown to be susceptible to JEV infection capable of developing lethal encephalitis, regardless of the route used for viral inoculation into cephalic cavities near to the brain, as well as at peripheral sites far from the brain. This suggests that JEV can reach the CNS via multiple pathways, replicate to a high titer within the CNS, and so cause neuropathological disease. The molecular basis of JEV neuroinvasion has not yet been fully understood and, thus, is an important area of research to better understand viral pathogenesis. 

#### 4.2.3. Peripheral Inoculation during Pregnancy 

While the risk of transplacental transmission of JEV from a pregnant woman to her unborn baby has not been conclusively determined [195,196,197,198], its occurrence and negative effects on the growing fetus have been demonstrated experimentally in mice [16,18,270,271,272,273]. In an early study, in which Swiss albino pregnant mice were inoculated intraperitoneally with a sublethal dose of JEV in the first week of gestation, the virus was detected in the placenta and fetal organs (e.g., brain, spleen, and liver), with the number of fetal and neonatal deaths estimated to be considerably higher in infected than uninfected mice [18]. The fetal and neonatal death rate was found to be dependent on the time when maternal infection occurred over the gestation period of ~21 days, as the highest death rate of 66% was obtained by the infection at the first week of gestation, followed by 49% at its second week, and 14% at its third week, compared to 3–7% of the uninfected control mice [18]. Apart from the fetal and neonatal deaths, however, no apparent congenital abnormalities were observed in the developing fetuses or newborns from JEV-infected mice [18]. A subsequent follow-up study has further shown that the transplacental transmission of JEV can occur repeatedly in two consecutive pregnancies six months apart, causing abortions more often, but neonatal deaths less frequently, during their second pregnancies than during their first [16]. Moreover, a series of previous small-scale experiments have suggested that the occurrence of transplacental JEV transmission in mice, as determined by detection of the virus in placentas and fetuses, varies with the strain of the mouse [270], strain of the virus [271], route of inoculation [272], and timing of the infection during pregnancy [273]. These available data collectively suggest that mice may serve as a small-animal model for the study of transplacental JEV transmission, which occurs in pigs and potentially in humans. Considering that other flaviviruses, such as WNV [274], ZIKV [275], and DENV [276,277] are shown to be able to pass through the placenta during pregnancy, there is a high likelihood of transplacental transmission of JEV in humans, capable of causing pregnancy loss and/or congenital anomalies. Further studies are desirable to assess the risk of transplacental JEV transmission in humans and its adverse effects on the development and survival of their fetuses. 

### 4.3. Viral Pathogenesis 

JEV causes a range of neurological complications [278,279], of which JE is of primary concern, as the virus can permanently impair the CNS via a complex multistep process starting with viral replication in the mononuclear phagocyte system, then viral entry into the CNS, and ending with viral replication in the CNS, which eventually results in neuronal cell death and detrimental neuroinflammation (Figure 4). This process is determined by viral pathogenicity referring to the following two main intrinsic properties [280,281]: (1) neuroinvasiveness, which represents viral ability to invade the CNS from the periphery, and (2) neurovirulence, which represents viral ability to cause pathological changes within the CNS. 

#### 4.3.1. Viral Replication in the Mononuclear Phagocyte System 

JEV gets into the body of vertebrate hosts, including humans, when an infected female mosquito bites their skin, under which the primary site of infection is the dermis, a layer of connective tissue that lies beneath the epidermis and contains lymph vessels, blood vessels, nerve endings, oil glands, sweat glands, and hair follicles [282]. At the biting site, the mosquito inserts her proboscis through the epidermis and into the dermis to reach a blood vessel and injects her saliva containing the virus into the blood stream before taking a blood meal [283]. During this blood intake process, JEV is believed to infect a subset of peripheral blood mononuclear cells (PBMCs) in humans, NHPs, and mice; namely mononuclear phagocytes (i.e., MΦs and DCs [217,224,284,285,286,287,288,289,290,291,292,293,294,295,296,297]) that play pivotal roles in initiating and orchestrating the host antiviral, inflammatory, innate and adaptive immune responses [298]. Other mosquito-borne flaviviruses, such as WNV [299,300,301,302], ZIKV [303], DENV [304,305,306], and YFV [307] are also shown to be able to infect mononuclear phagocytes. Although both MΦs and DCs are permissive for JEV replication, the replication efficiency depends on the cell type and the status of cell maturation. For instance, murine MΦs are found to be more permissive for JEV replication than murine DCs [287], and immature, but not mature, human DCs are shown to support productive JEV replication [289,290]. The initial JEV replication in the skin, together with the mosquito saliva deposited during blood feeding, is thought to trigger local inflammation, which promotes the recruitment of monocytes from the blood stream to the inflamed skin tissue, where they then differentiate into MΦs and DCs and become new targets for infection [282,308]. Although it is unclear if this is applicable for JEV, other flaviviruses may also target other immune (e.g., mast cells) and non-immune (e.g., keratinocytes and fibroblasts) cells in the skin [309,310,311]. Based on our current understanding, JEV-infected mononuclear phagocytes and cell-free JEV virions in the skin are believed to migrate through the lymph and blood vessels to peripheral lymphoid organs (e.g., lymph nodes and spleen), where JEV replication is shown to take place in mononuclear leukocytes, such as MΦs and, perhaps some T cells [299,312,313,314,315]. JEV may be able to replicate in human T cells at a very low rate and possibly in a strain-dependent manner [286,313]. 

The productive JEV replication in peripheral lymphoid organs causes an episode of transient viremia commonly at low levels [226,314,316,317,318], which may precede the initiation of systemic infection, the development of nonspecific symptoms, and the activation of antiviral immune systems, leading to the proliferation and differentiation of antigen-specific lymphocytes [319,320]. In murine systems, it has been shown that, upon infection with a WT JEV strain, MΦs induce the expression of costimulatory surface molecules (e.g., CD40, CD80, CD86, and MHC) and proinflammatory cytokines (e.g., IL-6 and IL-12) as part of the normal innate immune response [321,322], whereas DCs fail to upregulate some of those costimulatory surface molecules (e.g., CD40 and MHC), and intricately produce not only those proinflammatory cytokines but also some anti-inflammatory cytokines (e.g., IL-10) [284,287]. In the latter two studies, it has been further demonstrated that JEV-infected DCs are broadly inefficient in activating CD4^+^ and CD8^+^ T cells, notably with the key anti-inflammatory IL-10 produced by the infected DCs having a negative effect on their ability to activate CD8^+^ T cells, but not CD4^+^ T cells [284,287]. In line with these findings, both murine and human DCs, when infected with WT JEV, have been shown to expand a population of regulatory T (Treg) cells, which is mediated by the upregulation of PD-L1, a co-inhibitory molecule critical for the maintenance of immune homeostasis, on DCs [284,289]. Similar findings have been obtained in a recent transcriptomic study, showing that infection of human immature DCs with WT JEV increases the expression of proinflammatory cytokines and chemokines (e.g., IL-6, IL-12, TNF-α, CCL2, and CCL5), as well as the anti-inflammatory IL-10, and activates Treg cells [292]. These data thus suggest that JEV may interfere with the maturation of immature DCs, which leads to altered cytokine production, impaired T cell activation, and induced Treg cell expansion, thereby undermining the host innate and adaptive immune responses to the virus, and in consequence, allowing the virus to better replicate and disseminate throughout the host body. On the other hand, contradictory results have also been reported when mouse immature DCs are infected with the attenuated JEV vaccine strain SA_14_-14-2, which include the efficient maturation of immature DCs, high production of proinflammatory cytokines, effective activation of T cells, and impaired expansion of Treg cells [288]. Taken together, it suggests that JEV may alter the maturation and function of DCs, a professional antigen-presenting cell, following infection possibly in a strain-specific manner. A failure of the host to restrict JEV replication locally in peripheral tissues/organs, such as the skin, lymph nodes, and spleen, escalates the likelihood of the virus to spread throughout the host body to many different tissues/organs and eventually enter the CNS, before the adaptive immune responses are fully activated [323]. 

#### 4.3.2. Viral Entry into the Central Nervous System

Neurotropic viruses, in general, enter the CNS through the vascular or nervous system [324,325,326]. In the case of JEV, this neuroinvasion is believed to take place primarily through the vascular system, first, because the virus is amplified in peripheral lymphoid organs after the initial infection in the skin before it disseminates to many other organs including the CNS via a hematogenous route [26,248,327,328], and second, because its subsequent infection in the brain is quite diffuse and rife [329,330]. JEV is, therefore, considered to have the ability to enter the brain by crossing the blood–brain barrier (BBB), a highly selective physical barrier that prevents the influx of toxic substances, microbial pathogens, and immune cells to the brain parenchyma from the blood stream [331]. This function of the BBB depends largely on the tight junctions (TJs) between brain microvascular endothelial cells (BMECs) of the cerebral blood vessels that are ensheathed by pericytes and astrocyte foot processes [332,333,334,335]. Based on previous research, three possible mechanisms by which JEV can pass the BBB have been postulated [336,337,338]: 

(1) Transcellular transport of viral particles from one side of an endothelial cell to the opposite side through its interior [339]. A time-course electron microscopy analysis of brain sections of suckling mice after subcutaneous infection with JEV has suggested that viral particles are internalized into BMECs by both clathrin-dependent and clathrin-independent endocytosis, then transported across the cytoplasm by vesicle-mediated transcytosis, and eventually released into the brain parenchyma, with no visible damages at the BBB [340]. This study has also indicated that such a transcellular JEV transport may occur in pericytes as well [340]. Although JEV replication has not been consistently detected in vivo in BMECs of deceased JE patients or JEV-infected mice [209,327,329,341], because the virus may have been cleared at the time of analysis, it has been demonstrated in vitro by JEV infection experiments in cultured BMECs of humans and rats [342,343,344,345,346,347,348,349,350]. In these cultured BMECs, it is noted, though, that the level of JEV replication is unsustainably low and so has no or minimal effect on the cell viability, possibly due to the host antiviral/antiapoptotic responses [342,345,346,348,349]. Just like JEV, other flaviviruses, such as WNV and ZIKV, have also been shown to establish persistent infection with no obvious cytopathic effects (CPEs) in cultured human BMECs [351,352]. This lack of virus-induced CPEs therefore makes BMECs favorable for the transcellular transport of JEV, ZIKV, and perhaps other related neurotropic flaviviruses. In recent years, a high-throughput siRNA screen has found that a plasma membrane-actin cytoskeleton linker, ezrin, and its associated proteins are essential for the caveolae-mediated, clathrin-independent entry of JEV into human BMECs [344]. Moreover, it has been shown in human BMECs that JEV infection activates the epidermal growth factor receptor signaling pathway, which suppresses the antiviral IFN response and, in consequence, promotes viral replication [343]. Altogether, it is possible that JEV may traverse the BBB via a transcellular route, which involves viral entry at the apical surface of polarized BMECs and the release of those endocytosed virions or their progeny virions following viral replication, from the basolateral surface of the BMECs into the brain parenchyma without inducing host cell death. 

(2) Paracellular movement of viral particles through an intercellular space between two adjacent endothelial cells [353]. Mouse infection experiments have shown that JEV can enter the brain as the structural integrity of the BBB gets compromised due to the permeability-inducing effects of JEV-triggered inflammatory mediators, such as cytokines, chemokines, and matrix metalloproteases (MMPs) on BMECs [292,330,354,355,356,357,358,359]. In particular, multiple studies have suggested that proinflammatory cytokines (e.g., IL-1β, IL-6, and TNF-α) and chemokines (e.g., CCL2, CCL3, and CXCL2) play major roles in disintegrating the BBB in JEV-infected mice [354,356,359,360]. Similar cytokine- and chemokine-mediated BBB disruption is also implicated in JEV neuroinvasion in humans and NHPs [209,291,361]. The functional roles of cytokines and chemokines in JEV-induced BBB breakdown have been studied in in vitro co-culture systems: (1) A human BBB model using BMECs and astrocytes has shown that both cell types get infected with JEV and release proinflammatory cytokines and chemokines (e.g., IL-6, CCL5, and CXCL10), correlated with increased barrier permeability of BMEC monolayers [345]. (2) Rat BBB models using BMECs with either pericytes or astrocytes have demonstrated that proinflammatory cytokines (e.g., IL-6) released from JEV-infected pericytes and astrocytes activate the ubiquitin-proteasome system in BMECs for the degradation of cytoplasmic TJ proteins (e.g., zonula occluden-1 [ZO-1]) by upregulating the ubiquitin ligase UBR-1 through the activation of JAK-STAT signaling [348,349]. In all these BBB models, it is noted that both pericytes and astrocytes are susceptible to JEV infection, but neither shows any apparent JEV-induced CPEs. Aside from these in vitro BBB models, it has also been observed in non-brain endothelial and epithelial cells that JEV infection alters the expression of some important TJ proteins, which is sufficient to disrupt cell-cell junctions and thus to impair the barrier function of those cell monolayers [362]. Hence, it is likely that multiple cell types at the blood–brain interface may release proinflammatory cytokines and chemokines as an innate immune response to JEV infection, which then triggers the intracellular degradation of cytoplasmic TJ proteins in BMECs, thereby compromising the structural integrity of the BBB [338]. 

MMPs, a family of Zn^2+^- and Ca^2+^-dependent endopeptidases that are secreted by many cell types, including fibroblasts and leukocytes, into the extracellular milieu [363,364], are known to cleave the extracellular matrix as well as non-matrix proteins around the BBB [365,366,367,368]. In line with their function, MMPs, particularly MMP2 and MMP9, are found to be upregulated in the brain and serum of JEV-infected mice with increased BBB permeability [355,357,369]. Similarly, the involvement of MMPs in JEV-induced BBB breakdown is also indicated in humans and NHPs [291,370]. An in vitro human BBB model consisting of BMECs and astrocytes has shown that JEV infection induces MMPs (e.g., MMP7, MMP8, and MMP9), correlated with compromised barrier function of BMEC monolayers [345]. Likewise, an in vitro rat BBB model constituted with BMECs and astrocytes has revealed that the infection of astrocytes with JEV causes the release of not only IL-6 and vascular endothelial growth factor, which transduce signals to activate the ubiquitin-proteasome pathway for intracellular degradation of the cytoplasmic TJ protein ZO-1 in BMECs, but also MMP2 and MMP9, which are critical for extracellular degradation of the transmembrane TJ protein claudin-5 on the surface of BMECs [349]. In the rat astrocyte cell line RBA-1, it has also been shown by using pharmacological inhibitors and siRNAs that JEV infection induces the expression of MMP9, as a result of the activation of NF-κB and AP-1 through the generation of reactive oxygen species and activation of mitogen-activated protein kinases [371,372]. Moreover, a recent study using mast-cell-deficient Sash mice has demonstrated that mast cells are involved in BBB breakdown during JEV infection, and further, that chymase, a serine protease released from activated mast cells, has an active role in degrading not only transmembrane TJ proteins (claudin-5 and occludin), as consistent with its function, but also cytoplasmic TJ proteins (ZO-1 and ZO-2) by an unknown mechanism [373]. In addition, this study has further shown, using in vivo mouse and in vitro BBB models, that treatment of a chymase-specific inhibitor decreases BBB leakiness, reduces viral neuroinvasion, and thus lowers the morbidity and mortality associated with JE [373]. Since chymase converts the inactive zymogen form of MMPs, such as MMP2 and MMP9, into their enzymatically active forms [374], it may augment the role of MMPs in BBB disruption. However, one caveat of this study is that the mast-cell-dependent BBB breakdown was shown to be caused by infection with either a virulent WT strain (Nakayama) or an attenuated vaccine strain (SA_14_-14-2) of JEV, although the Nakayama strain produced significantly more severe neurological deficits and much higher mortality as compared to the SA_14_-14-2 strain [373]. Therefore, these data suggest that proteases, such as MMPs and chymases, are induced or activated during JEV infection and are directly involved in the extracellular degradation of transmembrane TJ proteins, thereby damaging the structural integrity of the BBB. 

Thus far, the accumulated evidence indicates that JEV may traverse the BBB via a paracellular route, expedited by at least two different types of inflammatory mediators: (1) cytokines and chemokines (e.g., IL-6 and CCL2), capable of inducing the ubiquitin-mediated proteasomal degradation of cytoplasmic TJ proteins in BMECs, and (2) extracellular proteases (e.g., MMPs and chymases), capable of directly degrading transmembrane TJ proteins on the cell surface of BMECs. Among other neurotropic flaviviruses, WNV has been shown to cause the degradation of several TJ proteins, correlated with increased levels of multiple MMPs in the brain, resulting in increased BBB permeability [375,376]. Considering that the structural integrity of the BBB requires the proper biogenesis of both cytoplasmic and transmembrane TJ proteins which is regulated at multiple levels, such as gene transcription, translation, post-translational modifications (e.g., phosphorylation and protein degradation) and subcellular localization [377,378], it will be interesting to determine whether and how JEV alters the biogenesis of TJ proteins, other than protein degradation, in BMECs. For the transmembrane TJ proteins that are expressed on the cell surface of BMECs and are, thus, susceptible to a variety of extracellular proteases, such as MMPs, it can be expected that the stability of those TJ proteins depends on the expression level of particular MMPs, the conversion rate from the inactive pro-MMPs to their active forms, and the presence and action of the tissue inhibitors of particular MMPs [357,379,380]. In addition, it will be worthwhile to investigate whether any innate immune cells other than mast cells contribute to BBB breakdown during JEV infection [381]. 

(3) Transmigration of virus-infected leukocytes through an intercellular space between two adjacent endothelial cells [382]. Physical breakdown of the BBB may contribute not only to the paracellular movement of viral particles, but also to the transmigration of infected leukocytes from the intravascular space into the brain parenchyma [330]. Accordingly, another possible mechanism for JEV neuroinvasion is believed to be mediated by peripheral leukocytes that are infected by the virus and so can act as “Trojan horses” to creep the virus through the BBB into the brain tissue [286,293]. In line with this hypothesis, the transendothelial leukocyte migration in the brain is well known to be promoted by a conglomerate of proinflammatory molecules, which mediate the recruitment and adhesion of circulating leukocytes to the endothelium and their extravasation and transmigration through the BBB, either directly or indirectly [383,384,385,386]. One or more such proinflammatory mediators are shown to be upregulated by JEV infection in primary cell cultures of human and rat BMECs, human DCs, and mouse MΦs, as well as in serum, spleen, and brain samples of mouse JE models, which include (1) cell adhesion molecules [387], such as intercellular adhesion molecule-1, vascular cell adhesion molecule-1, and platelet endothelial cell adhesion molecule-1 [318,342,345,355,388]; (2) cytokines and chemokines [389,390], such as IL-1β, IL-6, IL-12, IFN, TNF-α, CCL2, CCL3, CCL4, CCL5, CXCL1, CXCL2, and CXCL10 [287,292,293,318,342,345,348,349,354,356,360,388,391,392,393]; (3) MMPs [394,395], such as MMP2, MMP7, MMP8, and MMP9 [293,345,349,355,357,360]; (4) growth factors [396,397], such as vascular endothelial growth factor and granulocyte colony-stimulating factor [293,345,349]; and (5) reactive oxygen and nitrogen species [398,399], such as diatomic oxygen- and nitric-oxide-related compounds [293,355,358]. Similarly, higher levels of proinflammatory cytokines and chemokines are detected in the plasma and CSF of JE patients and JEV-infected NHPs [209,291,361]. Of the mononuclear phagocytes (e.g., MΦs and DCs) that are susceptible to JEV infection, MΦs are advantageous for transendothelial migration, because they support a higher level of viral replication than DCs, but unlike DCs, do not develop CPEs [286,287,293,294,329,360]. However, one caveat of this Trojan horse hypothesis is that JEV-infected mononuclear phagocytes have not yet been found unequivocally to infiltrate into the brain in humans and mice [209,293,329,400]. Moreover, it is documented that JEV infection of human endothelial cell lines leads to the production of proinflammatory cytokines (e.g., IFN-β and TNF-α), the induction of MHC molecules (e.g., HLA-A, -B, and -E), and the shedding of a soluble form of HLA-E that is generated by MMPs, although its roles in JEV infection are unknown [347]. Collectively, the trafficking of JEV-infected mononuclear phagocytes, or potentially other leukocytes, through the BBB into the brain is plausible but certainly needs further investigation. 

Despite substantial research efforts, the precise molecular mechanism underlying the entry of JEV into the CNS is still poorly understood. It is hypothesized that JEV may traverse the BBB to reach the brain parenchyma via a hematogenous route involving one or more of the following three possible mechanisms: (1) transcellular virus transport, (2) paracellular virus movement, and (3) transendothelial infected-cell migration. Of these, it is not yet clear which mechanism is responsible for JEV neuroinvasion. Recent data from a mouse model after intravenous inoculation of WT JEV have revealed that viral loads in the brain increase from day 2 after infection, along with elevated levels of proinflammatory mediators (i.e., cytokines and chemokines), but that BBB permeability remains unchanged until day 4 [318]. Subsequent in vitro endothelial permeability assays using a mouse BMEC cell line have shown that the structural integrity of BMEC monolayers is compromised not by direct infection of JEV but by brain extracts derived from JEV-infected mice [318]. Moreover, it has been demonstrated that the administration of a neutralizing antibody directed against IFN-γ, a potent immunoregulatory cytokine, alleviates BBB disruption in JEV-infected mice [318]. Overall, these results suggest that JEV is able to enter the brain prior to BBB breakdown, which is therefore a consequence of JEV neuroinvasion, rather than a prerequisite for it. Similar results have been observed in rodents with two other encephalitic flaviviruses, WNV [401] and tick-borne encephalitis virus [402]. Hence, it is conceivable that JEV may enter the brain by active transcellular transport across the intact BMEC at the early phase of neuroinvasion, but at the later phase, by the passive paracellular movement of viral particles and/or the sneaky transendothelial migration of virus-infected leukocytes through the intercellular space between the disrupted BMECs that had been caused by JEV-induced neuroinflammation in the brain parenchyma [403]. Recently, it has been suggested that JEV neuroinvasion is promoted by the nuclear protein HMGB1 secreted from BMECs that facilitates leukocyte transendothelial migration [404], but is restricted by the receptor tyrosine kinase AXL that controls the production of IL-1α, primarily from pyroptotic MΦs [405]. Whatever the underlying mechanism involved, the neuroinvasion of JEV into the CNS is an essential key step to start the sequence of events that lead to life-threatening encephalitis [210]. 

#### 4.3.3. Viral Replication in the Central Nervous System 

Soon after JEV gets into the CNS, the virus is detectable in the brain parenchyma and CSF of humans [317,327], NHPs [406], and mice [315]. Within their brains, neurons are the primary target cells of JEV, as they are not only highly susceptible to virus infection, but also highly permissive for virus replication [209,248,254,291,318,329,341,407]. Using an in vivo mouse model, combined with an in vitro mouse neurosphere culture system, JEV is shown to preferentially infect neural progenitor cells, which undergo cell cycle arrest, and is thus potentially capable of suppressing their proliferation and impacting neurogenesis [243,408]. In accordance with these findings, the susceptibility of rat neurons to JEV infection also appears to depend on the cell maturity status, both in vivo and in vitro, with the immature developing neurons determined to be most vulnerable [227,409]. In contrast to neurons, non-neuronal glia, such as astrocytes and microglia, are found to be rarely positive for viral antigens in JEV-infected humans [327,341], NHPs [254,291], and mice [318,330]. In in vitro cell cultures, on the other hand, it has been demonstrated that both astrocytes [345,349,409] and microglia [410,411,412] derived from humans and rodents are always susceptible to JEV infection, although its physiological relevance in vivo is unknown. Still, it is noteworthy that JEV infection in neurons is known to be strongly cytopathic, resulting in productive lytic infection in vivo, whereas JEV infection in microglia, if it occurs in vivo, is supposed to be noncytopathic or weakly cytopathic, possibly being capable of establishing persistent infection [410,411]. This raises a question whether microglia can serve as a long-term source of JEV in the CNS. This is an intriguing idea given the observation that JEV was detected in the CSF of a JE patient at ~4 months after initial infection [413], as well as in the PBMCs collected from some JE cases even after ~8 months of recovery [414]. In vitro, JEV is also reported to be able to establish persistent infection in human monocytes for >3 weeks [286]. Hence, these results suggest that JEV may persist in microglia, the MΦ-like immune cells important for the control of JEV replication in the CNS [415], thereby perhaps contributing to the poor clinical outcome of JE and the neurological sequelae affecting a significant portion of JE survivors. 

Based on our current understanding, the uncontrolled virus replication in neurons directly causes neuronal death in vivo in humans [327,329], NHPs [291], and mice [408,416], as well as in vitro in their neuronal cell cultures [286,408,417]. The neuronal death is accompanied by reactive gliosis involving the activation of glia such as astrocytes and microglia, which then release numerous inflammatory molecules as part of a defense mechanism that, paradoxically, give rise to multiple neuropathologic events, eventually aggravating both neurodegeneration and neuroinflammation, the two main clinical features of JE [26,209,243,291,318,407,416,418,419]. In this model, the major neuropathologic events occurring during JEV replication in the brain include: (1) *bystander neuronal death* [420,421], which is triggered by a variety of inflammatory mediators (e.g., IL-1β and TNF-α) released from JEV-activated glia, especially microglia, and also possibly from the JEV-infected/activated MΦs migrated from the periphery into the inflamed brain parenchyma [291,318,400,412,422,423,424,425]; (2) *neurogenesis impairment* [426,427], which is potentially caused not only by direct loss of neural stem/progenitor cells (NSPCs) due to JEV infection, but also by a mixture of inflammatory substances produced from JEV-activated microglia and possibly other glial cells that suppress the proliferation and differentiation of NSPCs [227,243,408,409]; (3) *BBB breakdown* [428,429], which is induced by at least two different types of inflammatory mediators released from glial, as well as non-glial, cells such as pericytes and mast cells: (*i*) cytokines and chemokines (e.g., IL-6 and CCL2), capable of triggering the intracellular degradation of cytoplasmic TJ proteins that is mediated by the ubiquitin-proteasome pathway in BMECs [209,291,318,345,348,349,354,356,359,360,361], and (*ii*) extracellular proteases (e.g., MMPs and chymases), capable of catalyzing the extracellular degradation of transmembrane TJ proteins on BMECs [291,345,349,355,357,369,370,371,372,373]; and (*4*) *leukocyte infiltration* [430,431], which is modulated by inflammatory molecules, such as cytokines, chemokines, and growth factors, released from JEV-activated microglia and other glial cells that regulate the trafficking of peripheral leukocytes (e.g., MΦs, neutrophils, and T cells) across the BBB into the inflamed brain parenchyma [243,291,315,329,330,355,360,407,423,432,433,434]. All these neuropathologic events are aggravated by amplification of both the production of inflammatory cytokines and chemokines [435] and the infiltration of leukocytes into the brain [422,436]. A host of previous studies have suggested that the intensity of neuroinflammation, featured by the secretion of proinflammatory cytokines, is associated with the level of viral load, impacting the clinical outcome of JE in both humans and experimental animals, although this notion remains debatable [26,153,209,223,224,291,329,358,361,400,419,424,433,437,438,439,440,441]. 

## 5. Conclusions and Perspectives

As a laboratory animal model that faithfully recapitulates human JE, mice have been used extensively for the study of JE neuropathogenesis, host response, and the safety and efficacy of new vaccines and therapeutics against JEV infection. It should be noted that the susceptibility of mice to JEV infection varies very much, depending not only on the viral strain, inoculum dose, and inoculation route employed, but also on the mouse strain and age used, although the underlying basis for the impact of host factors is largely undefined. Hence, the mouse strain and age-dependency should be taken into consideration for the design of future mouse infection experiments. 

Based on a large body of early and recent studies conducted primarily using a murine infection model, a working hypothesis of how JEV causes JE has been formulated: The transmission of mosquito-borne JE occurs when an infected female mosquito injects her saliva with JEV into the host’s skin for blood meals. Principally in the dermis, JEV first infects mononuclear phagocytes (i.e., MΦs and DCs), which in combination with the injected mosquito saliva, induces local inflammation that promotes the recruitment of monocytes from the blood to the inflamed skin tissue, where they differentiate into MΦs and DCs and become new targets for JEV infection. Thereafter, the infected cells and cell-free virions move through lymph and blood vessels to peripheral lymphoid organs (e.g., lymph nodes and spleen), where JEV replicates in mononuclear leukocytes (e.g., MΦs and possibly T cells). The JEV replication in peripheral lymphoid organs then results in transient viremia, which leads to a systemic infection, causing nonspecific symptoms and stimulating the immune system. If the innate immunity fails to control JEV replication in peripheral tissues/organs (e.g., skin, lymph nodes, and spleen), the virus is likely to spread throughout the body, including the CNS, before the adaptive immunity is fully developed. Occasionally, JEV crosses the BBB to get into the brain by one or more of the following mechanisms: transcellular virus transport, paracellular virus movement, and transendothelial infected leukocyte migration. Within the brain parenchyma, JEV replicates in neurons, which causes neuronal death and activates glial cells (e.g., astrocytes and microglia), releasing numerous inflammatory mediators responsible for bystander neuronal death, neurogenesis impairment, BBB breakdown, and leukocyte infiltration into the brain parenchyma. These neuropathologic events are associated with the worsening of both neurodegeneration and neuroinflammation, the two hallmarks of JE. 

Although significant progress has been made over the years in understanding the pathobiology of JEV infection, there are a number of outstanding questions that still need to be addressed in future studies, including: (1) What are the virus–host cell interactions required for viral entry into susceptible host cells, such as MΦs and neurons [137,138,139]? (2) What are the molecular and cellular mechanisms by which the virus enters the CNS from the periphery [336,337,338]? (3) What are the viral and cellular factors and their underlying mechanisms needed for viral neurovirulence within the CNS [103,442,443]? (4) What are the proinflammatory mediators and their underlying processes involved in promoting detrimental neuroinflammation [403,442,444]? These questions are anticipated to be answered in the upcoming years using a well-established murine model of JEV infection, combined with new innovative techniques in molecular biology, cell biology, and immunology, as well as powerful genetic tools for targeted genome engineering of both the virus and the host cell/organism. 

## Figures and Tables

**Figure 1 pathogens-12-00715-f001:**
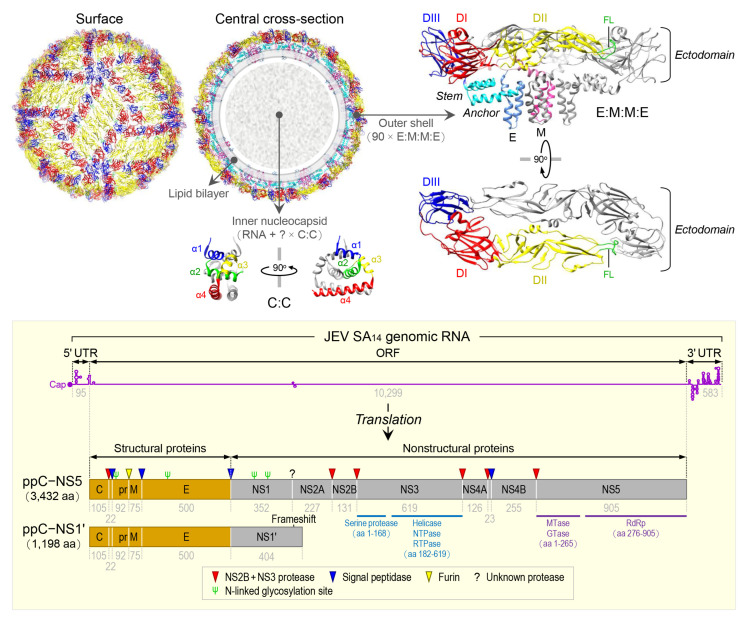
Virion structure, genome organization, and gene expression of JEV. The top panel shows the high-resolution cryo-electron microscopy structure of JEV strain P3 [102]. The virion contains multiple copies of three proteins: capsid (C), envelope (E), and membrane (M). The C monomer is a cytosolic protein containing four helices (α1, blue; α2, green; α3, yellow; and α4, red). The E monomer is an integral membrane protein comprising three topologically distinct parts: an N-terminal ectodomain, which has three structural domains (DI, red; DII, yellow; and DIII, blue) with the fusion loop (FL, green) positioned at the distal end of DII; a stem (cyan), which has three non-membrane-spanning helices; and a C-terminal anchor (cornflower blue), which has two membrane-spanning helices. The M monomer is also an integral membrane protein comprising three topologically distinct parts: an N-terminal extension containing an unstructured peptide fragment (pink), a non-membrane-spanning helix (hot pink), and a C-terminal anchor containing two membrane-spanning helices (deep pink). The bottom panel depicts the genome organization and gene expression of JEV strain SA_14_ [103]. The genome is a capped but unpolyadenylated plus-strand RNA, with a single long open reading frame (ORF) flanked by short, highly structured 5’ and 3’ untranslated regions (UTRs). The ORF is translated into two polyprotein precursors, both of which are cleaved by viral and cellular proteases, as indicated, to produce three structural (orange) and seven nonstructural (gray) proteins. Of these proteins, two are multifunctional enzymes: First, NS3 has serine protease, helicase, NTPase, and RTPase activities. Second, NS5 has MTase, GTase, and RdRp activities. Four N-linked glycosylation sites are marked, one in the pr portion of prM, one in E, and two in NS1/1’.

**Figure 2 pathogens-12-00715-f002:**
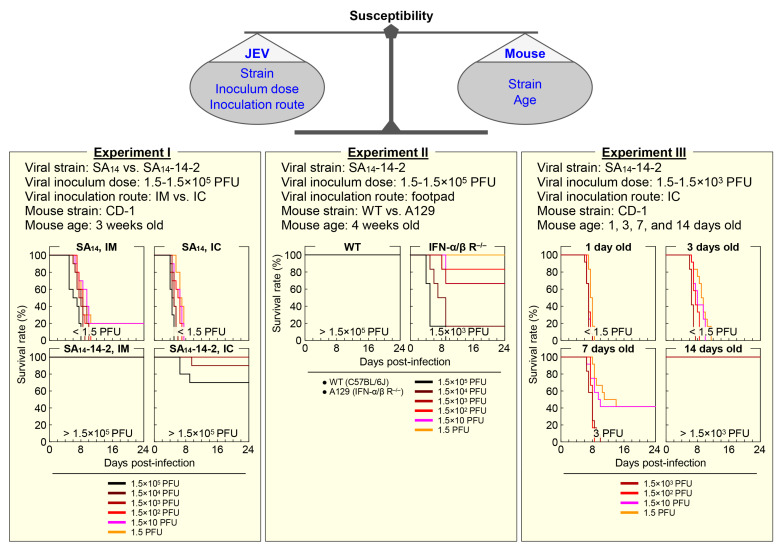
Variation in mouse susceptibility to JEV infection. Results from three mouse infection experiments are presented, demonstrating that the level of susceptibility of mice to JEV infection, primarily based on the mortality rate, is determined by multiple factors, such as the viral strain (SA_14_ vs. SA_14_-14-2), inoculum dose (1.5 to 1.5 × 10^5^ PFU/mouse), and inoculation route (intramuscularly [IM] vs. intracerebrally [IC]), as well as the mouse strain (wild-type [WT] vs. A129) and age (1, 3, 7, and 14 days). In each experiment, mortality was recorded for 24 days after infection. Experiments I and III are reproduced from our previous published report [103], and Experiment II is our unpublished data.

**Figure 3 pathogens-12-00715-f003:**
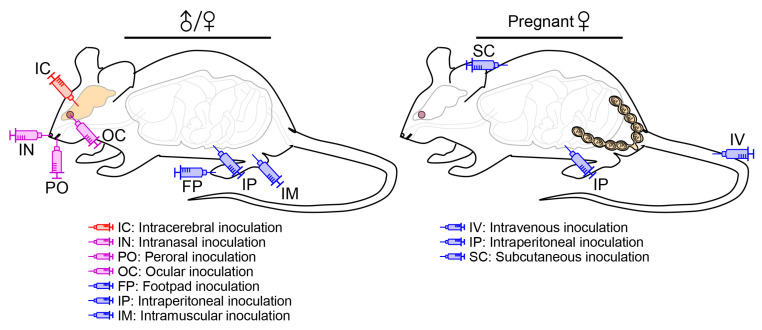
Inoculation routes for JEV infection in mice. Depending on the purpose of the research, experimental infection of mice with JEV has been performed by inoculating the virus directly into the brain (IC) or into a cephalic cavity, such as nasal (IN), oral (PO), and orbital (OC) cavities, or at an anatomic site distant from the brain, such as the hindlimb footpad (FP), peritoneal cavity (IP), and thigh muscle (IM). The transplacental transmission of JEV has been examined by inoculating the virus into the tail vein (IV) or the peritoneal cavity (IP) of pregnant mice, or under the loose skin (SC) of the back.

**Figure 4 pathogens-12-00715-f004:**
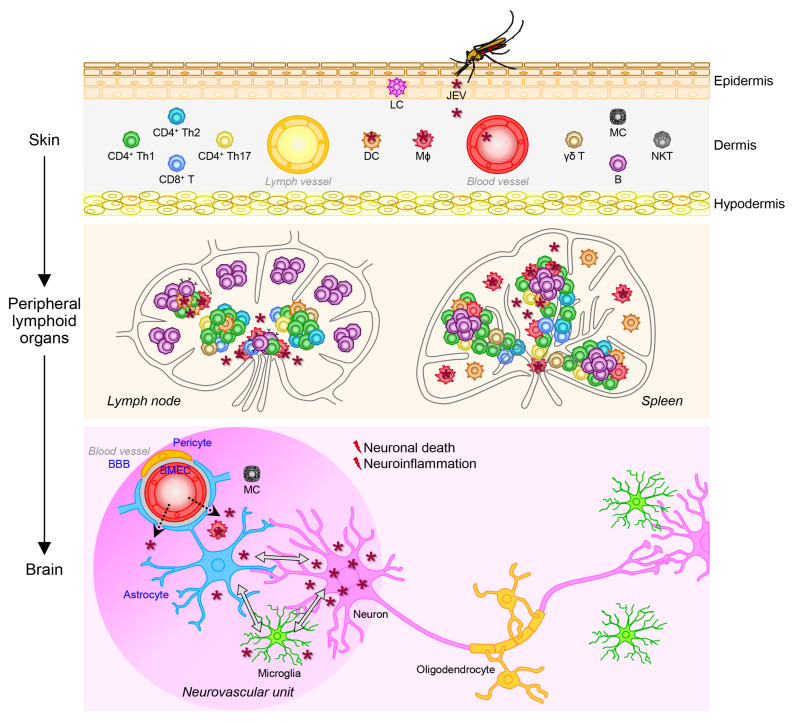
A working hypothesis for how JE, an acute inflammatory CNS disease, is caused by a mosquito-borne JEV infection. The infection begins with the bite of an infected mosquito that injects the virus from the salivary glands into the host’s skin, the dermis, where the virus presumably infects mononuclear phagocytes. From the dermis, the infected cells and cell-free virions may migrate through lymph and blood vessels to peripheral lymphoid organs, where the virus probably replicates in mononuclear leukocytes. Following viral amplification in peripheral lymphoid organs, JEV can spread into many other organs through the vascular system. In brains, JEV may cross the BBB to enter the brain parenchyma, where the virus replicates almost exclusively in neurons, causing neuronal death and activating glial cells. The activated glial cells then release numerous inflammatory mediators involved in various neuropathologic events, such as bystander neuronal death, neurogenesis impairment, BBB breakdown, and leukocyte infiltration. These neuropathologic events cause the worsening of both neurodegeneration and neuroinflammation.

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
