# Peer review of "Mice as an Animal Model for Japanese Encephalitis Virus Research: Mouse Susceptibility, Infection Route, and Viral Pathogenesis"

_pathogens, 2023, doi:10.3390/pathogens12050715_

Round 1
Reviewer 1 Report
The manuscript of J.C. Frank and co-authors is a very comprehensive and well-written review on using mice model for studying Japanese Encephalitis virus (JEV). My only suggestion regarding the content of the review will be to include a separate section that discusses the limitations of the mouse model. I also have to minor comments:
1. It is better not to use direct references to the figures (e.g. instead of “As illustrated in figure 1 [statement] use [statement] (Figure 1))
2. Figures seem to be of low resolution. Authors should include high quality images into the final manuscript.
Author Response
[REVIEWER #1’S COMMENTS]
The manuscript of J.C. Frank and co-authors is a very comprehensive and well-written review on using mice model for studying Japanese Encephalitis virus (JEV). My only suggestion regarding the content of the review will be to include a separate section that discusses the limitations of the mouse model.
>>> We are grateful for the reviewer’s comments. We have now further polished our original manuscript by discussing the limitations of the mouse model, as suggested by the reviewer. We now add: “However, there are also some limitations due to the fact that there exist a range of genetic, anatomical, physiological, and immunological differences between mice and humans [192,193], which reflects how well mice recapitulate key biological processes contributing to the development and progression of JEV-induced diseases. Indeed, a systematic comparative study has shown that in some human inflammatory conditions, the gene expression profiles are not reproduced in their mouse models [194]. Also, in cancer studies, it has been indicated that the process of tumorigenesis is fundamentally different in humans and mice [195] (Lines 248-255).”
Minor comments:
1. It is better not to use direct references to the figures (e.g. instead of “As illustrated in figure 1 [statement]” use [statement] (Figure 1)).
>>> As recommended by the reviewer, all four figure citations have now been presented in parenthesis (Lines 121, 122, 260, 344, and 498).
2. Figures seem to be of low resolution. Authors should include high quality images into the final manuscript.
>>> As requested by the reviewer, we have included high-quality images of our four figures in the revised manuscript (Lines 153, 263, 347, and 504).
Reviewer 2 Report
The manuscript is devoted to the comprehensive analysis of the Japanese encephalitis virus study including both current state and further development. All figures are excellent and will be used for generations to come.
Minor problem points are highlighted in yellow in the attached file.
General comments.
1. Flavivirus transmission can be carried out by multiple different ways such as horizontal and vertical transmission in both invertebrate and vertebrate hosts, sex transmission in mammals, non-viraemic transmission etc. It provides the stability of the whole system. Therefore, JEV does not depend on population dynamics of wild reservoir hosts species.
2. Linear laboratory mice lack the genetic diversity and environmental pressures characteristic of natural populations (Turner and Paterson, 2013). Comparison of wild adapted reservoir hosts with laboratory rodents for the JEV infection seems to be reasonable to estimate advantages and disadvantages of the models.
3. Conclusions are commonly based on confirmed data described in the manuscript with the corresponding references but not on personal suggestions and great expectations. The flavivirus system in wild nature with global climate changes is really complex and, therefore, is hard to predict.
Specific comments.
1. Figure 1 - the legend includes the repeat of the text from the previous paragraph and too long.
2. Line 117. What means "C:C homodimers"? Nothing was about the capsid protein and the corresponding abbreviation is missing.
3. Line 123. " poly(A) " at 3'-end. How about oligo(A) tracts in hypervariable region of 3'UTR with multiple INDELs?
4. Line 176. It is hardly possible to calculate "a median of 8.4 days". Miscalculations may be resulted from a limited sample size. The incubation period significantly varies and depends on both JEV inoculation dose, strain, innate and adaptive immunity of host etc.
5. Lines 283-284. "elicit stronger immune responses following peripheral JEV infection than C3H/He, which in turn suppress viral replication in the brain even if the virus enters the brain"
Immune responses are individual features. They vary among animals of the same line. Are the described differences statistically significant or not?
6. What means the abbreviation "MΦs"?

Author Response
[REVIEWER #2’S COMMENTS]
The manuscript is devoted to the comprehensive analysis of the Japanese encephalitis virus study including both current state and further development. All figures are excellent and will be used for generations to come. Minor problem points are highlighted in yellow in the attached file.
>>> We appreciate for the reviewer’s comments. We have now revised our original manuscript as suggested by the reviewer (as indicated below). All the modifications and additions we have made in the revised manuscript are marked in red.
General comments:
1. Flavivirus transmission can be carried out by multiple different ways such as horizontal and vertical transmission in both invertebrate and vertebrate hosts, sex transmission in mammals, non-viraemic transmission etc. It provides the stability of the whole system. Therefore, JEV does not depend on population dynamics of wild reservoir hosts species.
>>> The reviewer’s point is well taken. To make the description clearer, we have modified the sentence. We now say: “Of many susceptible animals, certain vertebrates such as suids (e.g., domestic pigs) and avians (e.g., wading birds) are especially relevant for the incidental transmission of JEV to the human population [8-10]” (Lines 40-42). Also, we have added one new sentence with appropriate references: “Experimentally, JEV can also be transmitted in the absence of mosquito vectors through several non-vector-borne routes in animals such as rodents, pigs, bats, and/or squirrel monkeys: contact transmission [11-14], aerosol transmission [11,15], transplacental transmission [16-20], and artificial insemination [21,22] (Lines 42-46).
2. Laboratory mice lack the genetic diversity and environmental pressures characteristic of natural populations (Turner and Paterson, 2013). Comparison of wild adapted reservoir hosts with laboratory rodents for the JEV infection seems to be reasonable to estimate advantages and disadvantages of the models.
>>> We appreciate for the reviewer’s comments. We have enhanced our original manuscript by adding a new sentence: “Moreover, it has been suggested that, in addition to laboratory mice, wild mice may offer some advantages in identifying cellular genes associated with individual differences in susceptibility to JEV infection within natural populations [224]” (Lines 308-311). The suggested reference has now been cited.
3. Conclusions are commonly based on confirmed data described in the manuscript with the corresponding references but not on personal suggestions and great expectations. The flavivirus system in wild nature with global climate changes is really complex and, therefore, is hard to predict.
>>> As suggested by the reviewer, we have modified the sentences. We now say: “Over the recent decade, however, the virus has expanded its territory to Europe and possibly Africa. This geographic expansion of JEV continues to pose a significant health threat to both people and livestock throughout the globe.” (Lines 834-836).
Specific comments:
1. Figure 1 - the legend includes the repeat of the text from the previous paragraph and too long.
>>> As suggested, we have removed the repetitive text and shortened the legend accordingly (Lines 154-172).
2. Line 117. What means "C:C homodimers"? Nothing was about the capsid protein and the corresponding abbreviation is missing.
>>> Thank you for picking up on this. The phrase has now been revised. We now say: “multiple capsid C:C homodimers” (Line 119).
3. Line 123. "poly(A)" at 3'-end. How about oligo(A) tracts in hypervariable region of 3'UTR with multiple INDELs?
>>> Yes, there is a variable region at the beginning of JEV 3’UTR, and its length varies due to a deletion of various sizes. However, this region is not essential for JEV RNA replication. Also, there are other highly conserved sequences and structures that are critical for viral replication. We feel that a detailed discussion on this topic is not within the scope of this paper.
4. Line 176. It is hardly possible to calculate "a median of 8.4 days". Miscalculations may be resulted from a limited sample size. The incubation period significantly varies and depends on both JEV inoculation dose, strain, innate and adaptive immunity of host etc.
>>> We agree. We have said: “… … it generally takes 5-15 days with a median of 8.4 days to show clinical signs/symptoms after contraction of the virus, but in some cases, may have a longer incubation time of 3-4 weeks [141]” (Lines 174-176). The related reference is cited.
5. Lines 283-284. "elicit stronger immune responses following peripheral JEV infection than C3H/He, which in turn suppress viral replication in the brain even if the virus enters the brain" Immune responses are individual features. They vary among animals of the same line. Are the described differences statistically significant or not?
>>> To address the question, we have revised the sentence. W now say: “Although the number of animals used in the study is small, these data suggest that … …” (Lines 284-285).
6. What means the abbreviation "MΦs"?
>>> Macrophages are abbreviated as MΦs. It is shown on line 291.
Reviewer 3 Report
I have found a previous systematic review with more comprehensive findings of the same topic.
https://journals.plos.org/plosntds/article?id=10.1371/journal.pntd.0010116
Author Response
[REVIEWER #3’S COMMENTS]
I have found a previous systematic review with more comprehensive findings of the same topic. https://journals.plos.org/plosntds/article?id=10.1371/journal.pntd.0010116
>>> We appreciate for the reviewer’s comments. This recent paper discussed about the susceptibility of mice to JEV infection, which is a small portion of our manuscript. In our manuscript, we have addressed key findings on mouse susceptibility, as well as the route of infection and viral pathogenesis. Also, we have further discussed some unanswered key questions for future studies.
>>> The suggested reference has now been cited (Lines 260-262).
Round 2
Reviewer 3 Report
The conclusions are a bit long, especially the contents from line 834 to 846 should be in the Introduction part only.
I personally do not understand why there are some red colour paragraphs of this revised version?
Author Response
[REVIEWER #3’S COMMENTS]
1. The conclusions are a bit long, especially the contents from line 834 to 846 should be in the Introduction part only.
>>> As suggested, we have now removed those sentences (lines 832-845) from the Conclusions and Perspectives section.
2. I personally do not understand why there are some red color paragraphs of this revised version?
>>> Those words and sentences in red were our modifications we made in the first revised manuscript. To avoid this confusion, the deletions (lines 832-845) made in this re-revised manuscript are now marked up using the “Track Changes” function.